# IL33/ST2 Axis in Diabetic Kidney Disease: A Literature Review

**DOI:** 10.3390/medicina55020050

**Published:** 2019-02-14

**Authors:** Alessandro Tonacci, Paolina Quattrocchi, Sebastiano Gangemi

**Affiliations:** 1National Research Council of Italy, Institute of Clinical Physiology (CNR-IFC), Via Moruzzi 1, 56124 Pisa, Italy; atonacci@ifc.cnr.it; 2School and Operative Unit of Allergy and Clinical Immunology, Department of Clinical and Experimental Medicine, University of Messina, Via Consolare Valeria, 98125 Messina, Italy; pquattrocchi@unime.it

**Keywords:** alarmin, chronic kidney disease, diabetes mellitus, IL-33, renal disease, ST2

## Abstract

Interleukin-33 (IL-33) is a cytokine belonging to the IL-1 family, playing a role in inflammatory, infectious and autoimmune diseases and expressed in the cellular nucleus in several tissues. High levels of IL-33 are expressed in epithelial barrier tissues and endothelial barriers. ST2 is a receptor for IL-33, expressed selectively on a subset of Th2 cells, mediating some of their functions. The IL-33/ST2 axis plays an important role in several acute and chronic inflammatory diseases, including asthma and rheumatoid arthritis. Different disorders are related to the activity of IL-33, ST2, or their axis, including cardiovascular disease or renal disturbances. Therefore, in the present work, a literature review was conducted, covering the period from 1 January 2000 to 30 November 2018, in PubMed, ScienceDirect, and Google Scholar database, to assess the involvement of the IL-33/ST2 axis in diabetic kidney disease. 6 articles directly dealing with the argument were identified, highlighting a clear link between IL-33/ST2 axis and diabetic kidney disease or related nephropathy. Overall, the involvement of ST2 seems to be more predictive than IL-33, especially in investigating the deterioration of kidney function; however, both compounds are pivotal in the field of renal diseases. Future studies are required to confirm the scientific evidences on larger and more heterogeneous cohorts.

## 1. Introduction

Interleukin-33 (IL-33) represents a recently discovered cytokine belonging to the IL-1 family, and plays a major role in inflammatory, infectious, and autoimmune diseases [1]. It is expressed as a nuclear alarmin in the nucleus of endothelial cells, fibroblasts, and epithelial cells in several tissues. Its role as an alarmin is pivotal in individuating damages in several inflammatory conditions, including atopic dermatitis and skin diseases [2,3]. It is released to the extracellular space in response to mechanical stress or cellular injury, and it elicits inflammatory responses. High levels of IL-33 are expressed in epithelial barrier tissues and endothelial barriers [4,5]. ST2 is a receptor for IL-33 and a member of the overall IL-1 receptor family. It is expressed selectively by many immune cells involved in type-2 immune response, including the group 2 innate lymphoid cells (ILC2s), mast cells, Th2 cells, eosinophils, basophils, as well as dendritic cells [6,7]. ILC2 recruitment is strongly related to the activity of IL-33, with their role being central in immunity against pathogens, type 2 inflammation, and tissue homeostasis and repair [8]. In this domain, IL-33 induces IL-5 and IL-13 release by ILC2, playing a key-role in the framework of allergic inflammation, including asthma and atopic dermatitis [9,10].

When it bonds with the ST2 receptor, IL-33 promotes the activation of nuclear factor (NF)-κB and mitogen-activated protein kinase, leading to an increased transcription of Th2 cytokines [11]. Therefore, it appears evident that the IL-33/ST2 axis mainly induces Th2 cytokines, differently from what occurs with other IL-1 family cytokines. Overall, this axis plays a key-role in several acute and chronic inflammatory diseases [12,13,14], including asthma [7] and rheumatoid arthritis [15].

In a human, IL-33 is expressed in human secondary lymphoid tissues, including the lymph nodes and appendix, and is spread along the vascular tree (large and small blood vessels from normal tissues, including the liver, skeletal muscle, kidney, and prostate, despite the micro-circulation of the brain and kidney glomeruli) [16]. Several literature works have highlighted the broad expression of IL-33 in normal, tumor, and chronically inflamed human tissues [16,17], confirming their significant expression in the kidneys [18,19].

Overall, apparently different disorders are related to the activity of IL-33, ST2 or their axis, including cardiovascular diseases [20], or renal disturbances, like chronic kidney disease (CKD), an immune inflammatory disease whose function is related to the presence of inflammatory biomarkers as much as the kidney function declines [21,22] (Figure 1).

So far, it is well known that diabetes is closely related to kidney disorders, being the most common cause of end-stage renal disease [23], with hyperglycemia-induced hemodynamic and metabolic pathways as the mediators of kidney injury [24].

Given the widespread diffusion of renal diseases and diabetes worldwide and, consequently, the high impact such diseases have on society and on healthcare systems, we performed a literature review about the IL-33/ST2 axis involvement in diabetic kidney disease or related nephropathy. This review is mainly focused on the pathophysiological outcome of the mentioned axis in the relevant disorders.

## 2. Studies on the IL-33/ST2 Axis Involvement in Diabetic Kidney Disease or Related Nephropathy

This literature review has retrieved a handful of articles directly dealing with the IL-33/ST2 axis involvement in diabetic kidney disease or related nephropathy. The articles taken into account are displayed in Table 1.

### 2.1. Results Achieved by the Studies on CKD

One of the first studies on the argument was published in 2012 by Bao and colleagues [25]. Their focus was the investigation of the serum levels of IL-33 and sST2 in patients with CKD, a sub-group of which were diabetic, and their association with disease severity. Based on the eGFR, the patients were classified into five categories of disease severity, plus a cohort of healthy individuals. sST2 was seen to be higher among CKD patients with respect to controls (*p* = 0.003), whereas no difference was seen for IL-33 (*p* = 0.656). By investigating the relationship between sST2 (and IL-33) and disease stage, the authors found a significant correlation between the serum level of sST2 and disease severity (*r* = 0.586; *p* < 0.001).

A larger cohort (238 patients with CKD, of which 60 had diabetes) was enrolled in an observational study conducted in Turkey some years later [28]. Here, serum IL-33/ST2 levels were investigated together with their association with cardiovascular problems. In this regard, Flow-mediated dilatation (FMD) was employed as a marker for endothelial function (see [31] for details). IL-33 and ST2 levels were increased as eGFR decreased, whereas the FMD was seen to be lower in more severe CKD patients. Overall, both IL-33 and ST2 were inversely associated with FMD, with ST2 being being a reliable predictor. Finally, both IL-33 and ST2 were associated with the risk of cardiovascular events, with higher cumulative survival in patients with lower IL-33 and ST2, compared with patients with higher values of such compounds. Such risks were also seen to be associated with Diabetes Mellitus, smoking, and proteinuria.

### 2.2. Results Achieved by the Studies on Kidney Diseases-Related Diabetes

As widely known, one of the most important co-morbidities of renal diseases is diabetes. In fact, for example, Type-2 diabetic nephropathy (DN) is considered the most common cause of end-stage renal disease among patients undergoing renal dialysis [32,33]. Some of the works retrieved in the present review have analyzed diabetes related to renal diseases, starting from the research by Caner and colleagues [26].

In that study, it was investigated whether IL-33 can be used to show the early stage of kidney injury in diabetic patients. To accomplish this aim, 42 patients with DM with normal renal function, 32 patients with DM + MA, and 26 healthy controls were enrolled and assessed in terms of IL-33 levels. The IL-33 levels were higher overall in DM + MA, followed by DM and controls. The differences between DM + MA and DM were not significant; however, a statistically significant difference was seen for both groups with respect to controls.

Later on, Shruthi and colleagues [27] investigated the serum levels of several cytokines, including IL-33 in 125 T1DM subjects, 96 of which featured microvascular complications (retinopathy and nephropathy in particular) and 29 without such concurrent problems, as well as in 76 controls. All the cytokines studied, including IL-33, were higher in T1DM subjects, but without displaying any differences between subjects with, and without, microvascular complications (MVC).

Samuelsson et al. [29], in a prospective cohort study, tried to investigate the plasma levels of sST2 and to determine whether sST2 could be a reliable early predictor for diabetic complications (including nephropathy and retinopathy) in young (15–34 years old) patients with DM. For this study, 220 patients were enrolled and, after a 10-year follow-up period, 112 of them developed diabetes-related complications, including retinopathy (*n* = 91), nephropathy (*n* = 12) or both of them (*n* = 9). At the time of diagnosis, plasma levels of sST2 were significantly higher in patients who later developed nephropathy with respect to the others.

### 2.3. Results Achieved by the Studies on Other Renal Diseases

End-Sstage renal disease (ESRD) was investigated twice by the same group [30,34], with the second study making use of part of the data collected during the first one. In the study published in 2016, 55 patients with ESRD were assessed in terms of sST2, evaluated by ELISA, before and after Hemodiafiltration, showing that sST2 was unaffected by the treatment cited. In 2018, the cohort was increased up to 123 patients, who were prospectively followed up from the date of the sST2/NT-proBNP measurement until their death or maximally up to 829 days. Those patients were divided into low- (<35 ng/mL) and high- (≥35 ng/mL) sST2 level cohort, according to the concentration of sST2 measured at the start of the study. The survived patients (89 out of 123 patients) showed a lower sST2 level with respect to deceased subjects, with the latter showing a significantly lower survival rate (*p* < 0.01) according to the Kaplan-Meier survival analysis. The analysis performed demonstrated that sST2 alone, and in combination with NT-proBNP, can predict the causes of mortality, and is independent from both renal function and hemodiafiltration treatment.

## 3. IL-33/ST2 Axis Involvement in Diabetic Kidney Disease or Related Nephropathy 

### 3.1. IL-33/ST2 in Diabetic Kidney Disease

IL-33 was first identified as a member of the IL-1 family in 2005 [7], and was seen as acting as a ligand for the ST2. As stated, the ST2, a receptor for IL-33, is selectively expressed by several immune cells involved in type-2 immune response, like ILC2 cells. The cells are recruited through the activity of IL-33, and are useful in the immunity against pathogens, type 2 inflammation, tissue homeostasis, and repair [8], and pivotal in conditions like asthma and atopic dermatitis [9,10].

According to the literature, sST2 serum levels were higher in CKD and associated with the severity of the disease [25], possibly having a role in the development of CKD or as a marker of disease severity. The role for sST2 appears to be stronger than the role of IL-33, as demonstrated by the lack of association between the increased IL-33 levels and kidney injury in diabetic nephropathy, with such increase possibly associated with diabetic disease [26]. Taken together with the evidences of other studies, which are not included in the detailed review as they do not directly deal with the conditions investigated in this work [35,36], such retrievals suggest that the serum levels of sST2 are more relevant in investigating the deterioration of kidney function with respect to the serum levels of IL-33.

As stated, diabetic nephropathy (DN) is probably the most prevalent etiology for CKD, featuring increases in excretion of urinary albumin, elevated blood pressure coupled with glomerular lesions, and functional loss of glomerular filtration leading to renal failure [37]. During its progression, several evidences for inflammatory responses have been shown, therefore the involvement of inflammatory cytokines in disease progression and severity is clear. The previously reported study by Caner and colleagues [26] showed that IL-33 levels were higher in DM patients with associated MA, followed by DM patients and controls. This demonstrated the up-regulation of IL-33 in diabetes, but failed to reliably show the usefulness of IL-33 for early diagnosis of DN. Interestingly, IL-33 was also significantly elevated in T1DM subjects, but without significant differences between those with, and without, microvascular complications [27]. On the other hand, the involvement of the ST2 expression was clear in all studies published to date, making it a reliable factor alone, or taken together, with IL-33, for the involvement in diabetic complications [29,38,39].

### 3.2. IL-33/ST2 in Other Renal Diseases

IL-33 and ST2 were also investigated in renal cell carcinoma (RCC) by Wu et al. [40]. A significant correlation between IL-33 expression and metastasis stage was found, and obviously this was also related to the overall prognosis of the patients. Indeed, the more advanced the metastasis stage, the worse the prognosis, and the higher the expression of IL-33. It was then hypothesized that IL-33 promotes RCC cell proliferation and chemotherapy resistance via its receptor ST2 and the JNK signaling activation in tumor cells. Therefore, targeting IL-33/ST2 and JNK signaling may have potential value in the treatment of RCC. However, the results obtained should be further confirmed in future studies on larger cohorts.

Another role for IL-33 was demonstrated by Akcay and colleagues [41], finding that IL-33 promotes acute kidney injury (AKI), through CD4 T cell-mediated production of the pro-inflammatory chemokine CXCL1. The inhibition of either IL-33, or CXCL1, might have a significant therapeutic potential in AKI. Therefore, studies dealing with the targeting of IL-33 or IL-33/ST2 axis should be carried out in order to verify their effective usefulness in decreasing the burden of Diabetic kidney disease (DKD) and other renal diseases, where a role for this axis was hypothesized, as already occurring in other conditions, including inflammatory diseases [42,43].

Finally, IL-33 could be a factor in renal transplantation, since a serum up-regulation of this compound would contribute to the pathogenesis of the chronic allograft dysfunction (CAD), endangering the long-term allograft survival in in kidney transplant recipients (KDR) [44]. In such patients, higher circulating IL-33 is associated with diminished glomerular filtration rate, age, diabetes, serum phosphorus and microalbuminuria, as well as with higher score for major adverse cardiovascular events, supporting a role for IL-33 in the cardiovascular burden for KDR [45]. End-stage renal disease was studied in two consecutive studies conducted by Homsak and Ekart [30,34]. In light of their findings, sST2 was demonstrated to have a high prognostic value, independent of renal function and of hemodiafiltration treatment. It was demonstrated to be a useful prognostic marker for stratifying ESRD patients on HDF, at a high risk for life-threatening events, hospitalization, and death, especially in combination with NT-proBNP. Furthermore, sST2 levels were seen to be correlated with disease activity in systemic lupus erythematosus (SLE), thereby suggesting a potential role for this compound as a marker for disease activity [46].

## 4. Conclusions

Despite the relatively low number of studies published to date, the link between IL-33/ST2 axis and diabetic kidney disease, appears to be more clear, with significant correlations with the disease stage. The ST2 involvement appears to be more predictive than IL-33, especially in investigating the deterioration of kidney function; however, both compounds are pivotal in this topic, with future studies confirming the scientific evidences on larger and more heterogeneous cohorts.

## Figures and Tables

**Figure 1 medicina-55-00050-f001:**
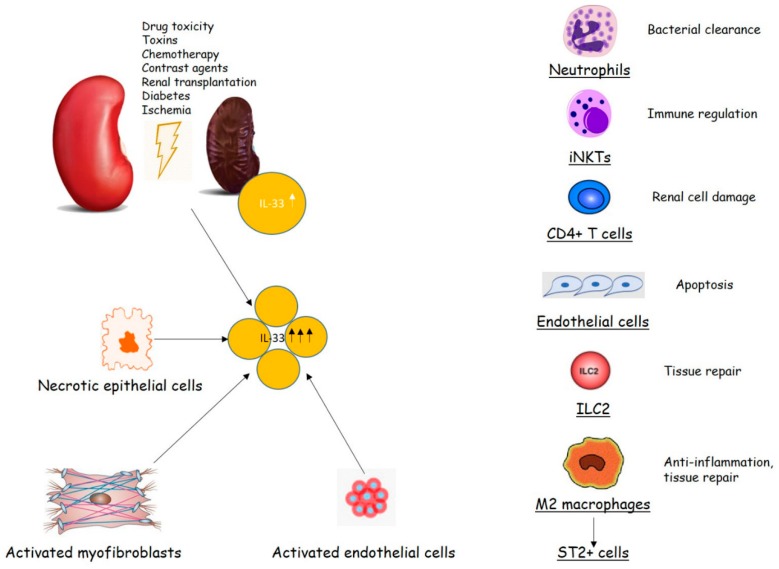
Interleukin-33 (IL-33)/ST2 signaling in renal injury (modified from [5]).

**Table 1 medicina-55-00050-t001:** Study selection.

Study	N (Case/Controls)	Design	Subject Characteristics	Findings
Bao et al. (2012) [25]	81 (69/12)	81 subjects (69 CKD divided for stage groups based on eGFR, 12 Con). Serum IL-33 and sST2 levels were estimated by ELISA	Mean Age: 48 years (range 19–72) Gender: 44 M, 25 F Origin: Chinese	No difference in IL-33 concentration between CKD and Con. Higher sST2 in CKD, correlated with disease severity
		9 DKD patients		
Caner et al. (2014) [26]	100 (42/32/26)	100 subjects (42 DM, 32 DM+MA, 26 Con). IL-33 investigated for showing the feasibility to retrieve the early stage of kidney injury in DM. Peripheral blood drawn, centrifuged, stored and analyzed by ELISA	Mean Age: 55.3 ± 12.4 years Gender: 40 M, 60 F Origin: Turkish	IL-33 higher among DM+MA, followed by DM and Con. Con had lower IL-33 concentrations than both DM+MA and DM. No difference between DM+MA and DM
		74 patients with DKD (32 of which with associated MA)		
Shruthi et al. (2016) [27]	201 (125/76)	125 T1DM, of which 96 with MVC. Serum levels of IL-33 calculated by ELISA	Mean Age: 21.5 ± 11.0 (T1DM), 29.0 ± 15.2 (T1DM + MVC), 32.0 ± 1.7 (Con) Gender: 15 M, 14 F (T1DM), 53 M, 43 F (T1DM + MVC), 31 M, 65 F (Con) Origin: Indian	IL-33 significantly elevated in T1DM subjects. No significant difference between those with and without MVC
		125 patients with DKD (29 without MVC, 96 with MVC)		
Gungor et al. (2017) [28]	238 (238/0)	238 patients with different stages of CKD based on eGFR. Serum IL-33 and sST2 evaluated with ELISA. Correlations searched with vascular damage and CV events	Mean Age: 48 years (range 26–69) Stage 1 CKD, 53 years (range 28-67) Stage 2 CKD, 49 years (range 27–69) Stage 3 CKD, 49 years (range 29–69) Stage 4 CKD, 49 years (range 26–69) Stage 5 CKD Origin: Turkish	IL-33 and ST2 inversely correlated with eGFR and associated with FMD. ST2 predictor of FMD. DM, smoking, proteinuria, IL-33, and ST2 associated with the risk of CV events
		60 patients with DKD (8 with Stage 1 CKD, 13 with Stage 2 CKD, 11 with Stage 3 CKD, 13 with Stage 4 CKD, 15 with Stage 5 CKD)		
Samuelsson et al. (2017) [29]	220 (220/0)	220 DM young patients enrolled. Prospective study on their outcome and relationship with plasma sST2 levels on different complications groups	Mean Age: 26.7 ± 6.1 (DKD) 25.0 ± 5.5 (no DKD) Gender: 15 M, 6 F (DKD), 118 M, 81 F (no DKD) Origin: Swedish	Plasma levels of sST2 significantly higher in *n* = 21 patients who later developed nephropathy compared to those *n* = 199 who did not
		220 patients with DM, of which 21 developed DKD 10 years later		
Homsak and Ekart (2018) [30]	123 (123/0)	123 ESRD patients divided by high and low sST2, checked by ELISA. Survival Ratio calculated	Mean Age: 66 years (range 25–88) Gender: 72 M, 51 F Origin: Slovenian	sST2 has a prognostic value, independent of renal function or HDF treatment
		40 patients with DKD		

CKD: Chronic kidney disease; Con: Controls; CV: Cardiovascular; DKD: Diabetic kidney disease; DM: Diabetes mellitus; eGFR: estimated glomerular filtration rate; ELISA: Enzyme-linked immune-Sorbent assay; ESRD: End-stage renal disease; FMD: Flow-mediated dilatation; HDF: Hemodiafiltration; IL-33: Interleukin-33; JNK: c-Jun N-terminal kinase; MA: Microalbuminuria; MVC: Microvascular complications; RCC: Renal cell carcinoma; sST2: Soluble ST2; T1DM: Type-1 diabetes mellitus; TNM: tumor-lymph node-metastasis.

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
