# Peer review of "IL33/ST2 Axis in Diabetic Kidney Disease: A Literature Review"

_medicina, 2019, doi:10.3390/medicina55020050_

Round 1

Reviewer 1 Report

Thank you for submitting the manuscript to 'Medicina.' 

Tonacci et al. performed a literature review on the IL-33/ST2 axis in renal disease and diabetes.

The authors have done a decent job in planning the review and preparing the manuscript.

Here are my comments and suggestions to the authors,

A. Major comments:

- The title and abstract are both confusing,

The objective not explicitly mentioned in the abstract. The authors wrote ‘Different disorders are related to the activity of IL-33, ST2 or their axis, including cardiovascular diseases or renal disturbances. A literature review was then conducted, covering the period from 1 January 2000 to 30 November 2018, in PubMed, ScienceDirect, and Google Scholar database’.

They should be clearer if they are studying the IL33/ST2 axis in diabetic kidney disease vs. others. Need to modify the title for the same reason. Can say ‘IL33/ST2 axis in Diabetic kidney disease’.

-Materials and methods,

Of the eight studies the authors included in the article, the studies by Caner et al. (2014), Shruthi et al., (2016) and Samuelson et al. (2017) dealt with diabetic kidney disease or diabetes patients. Others are not as, Wu et al., study was on renal cell carcinoma patients and Homsak et al., done two studies on the ESRD patients. And, interestingly in the study population in Bao et al., only 9 out of 69 patients (13%) had diabetic nephropathy, and the majority were with chronic Glomerulonephritis. In Gungor’s study, 60 out of 238 patients had Diabetic kidney disease. Why did the authors include the eight instead of the above initial three studies? As the authors wrote in the introduction ‘we performed a literature review about the IL-33/ST2 axis involvement in diabetic kidney disease or related nephropathy’.

-Discussion,

Not sure the need for 4th paragraph discussing the IL33/ST2 axis in renal cell carcinoma as the current review is about diabetic kidney disease. The entire study needs to be omitted unless the authors want to discuss in ‘IL33/ST2 and other kidney diseases’ section.

Same applies to 5th paragraph.

-Conclusions,

The authors wrote ‘Despite the relatively low number of studies published to date, the link between IL-33/ST2 axis and renal as well as diabetic kidney disease appears to be more and more clear’. Very confusing. What do they mean by renal as well as diabetic kidney disease?

I suggest the authors have the manuscript reviewed by a nephrologist for proper language and corrections.

B. Minor comments:

Introduction:

Discuss the distribution of IL33/ST2 in Kidney. And if the literature is available to discuss the location of the IL33/T2 axis in the renal tissue and pathological involvement of the diabetic kidney disease. For example, Chen et al. found that expression of ST2 is upregulated in human diabetic kidney tissues, (5).

Page 5 of 8.

Line 98. Authors wrote ‘A larger cohort (238 patients with CKD) was enrolled in an observational study conducted in Turkey some years later [28]. Reference 28 is wrong. Should be 25.

Page 5 of 8,

Line 131, Under Studies on other renal diseases:

Consider discussing, the study by Mok et al., (1) on the IL33/ST2 axis in SLE and Lupus nephritis. Authors found that elevated serum ST2 level did not discriminate between active lupus nephritis and non-renal lupus. And about the IL33/ST2 in Acute kidney injury, (2) and in renal transplantation, (3,4).

References:

1. Mok, M.Y.; Huang, F.P.; Ip, W.K.; Lo, Y.; Wong, F.Y.; Chan, E.Y.; Lam, K.F.; Xu, D. Serum levels of IL-33 and soluble ST2 and their association with disease activity in systemic lupus erythematosus. Rheumatology 2010, 49, 520–527

2. Akcay, A.; Nguyen, Q.; He, Z.; Turkmen, K.; Won Lee, D.; Hernando, A.A.; Altmann, C.; Toker, A.; Pacic, A.; Ljubanovic, D.G.; et al. IL-33 exacerbates acute kidney injury. J. Am. Soc. Nephrol. 2011, 22, 2057–2067.

3. Zhang, J.; Wang, Z.; Xu, Z.; Han, Z.; Tao, J.; Lu, P.; Huang, Z.; Zhou, W.; Zhao, C.; Tan, R.; et al. The potential role of IL-33 in renal transplant recipients with chronic allograft dysfunction. Ann. Transpl. 2016, 21, 611–618.

4. Mansell, H.; Soliman, M.; Elmoselhi, H.; Shoker, A. Elevated circulating Interleukin 33 levels in stable renal transplant recipients at high risk for cardiovascular events. PLoS ONE 2015, 10, e0142141.

5. Chen, N.K.; Chong, T.W.; Loh, H.L.; Lim, K.H.; Gan, V.H.; Wang, M.; Kon, O.L. Negative regulatory responses to metabolically triggered inflammation impair renal epithelial immunity in diabetes mellitus. J. Mol. Med. 2013, 91, 587–598.

Author Response

Thank you for submitting the manuscript to 'Medicina.'

Tonacci et al. performed a literature review on the IL-33/ST2 axis in renal disease and diabetes.

The authors have done a decent job in planning the review and preparing the manuscript.

Here are my comments and suggestions to the authors,

A. Major comments:

- The title and abstract are both confusing,

The objective not explicitly mentioned in the abstract. The authors wrote ‘Different disorders are related to the activity of IL-33, ST2 or their axis, including cardiovascular diseases or renal disturbances. A literature review was then conducted, covering the period from 1 January 2000 to 30 November 2018, in PubMed, ScienceDirect, and Google Scholar database’.

Thank You. We agree with Your point and we modified the relevant sentence in the Abstract as “A literature review was then conducted, covering the period from 1 January 2000 to 30 November 2018, in PubMed, ScienceDirect, and Google Scholar database, to assess the involvement of the IL-33/ST2 axis in Diabetic kidney disease.

They should be clearer if they are studying the IL33/ST2 axis in diabetic kidney disease vs. others. Need to modify the title for the same reason. Can say ‘IL33/ST2 axis in Diabetic kidney disease’.

Thank You for Your observation. We accepted Your suggestion about the title and modified it accordingly.

-Materials and methods,

Of the eight studies the authors included in the article, the studies by Caner et al. (2014), Shruthi et al., (2016) and Samuelson et al. (2017) dealt with diabetic kidney disease or diabetes patients. Others are not as, Wu et al., study was on renal cell carcinoma patients and Homsak et al., done two studies on the ESRD patients. And, interestingly in the study population in Bao et al., only 9 out of 69 patients (13%) had diabetic nephropathy, and the majority were with chronic Glomerulonephritis. In Gungor’s study, 60 out of 238 patients had Diabetic kidney disease. Why did the authors include the eight instead of the above initial three studies? As the authors wrote in the introduction ‘we performed a literature review about the IL-33/ST2 axis involvement in diabetic kidney disease or related nephropathy’.

Thank You for Your comment. We tried to include as many studies as possible, dealing with the topic, or potentially useful for the relevant discussion. However, as You correctly observed, there were some studies, which are significant to be discussed, that did not include diabetic patients (or where the inclusion of diabetic patients was not specified). Such studies were removed from the Table and discussed in comparison with the results obtained on patients with DKD. In addition, within Table 2, we added a line citing, for all the studies included in the review, the number of patients with DKD, in order to provide the reader with a more precise estimate of this amount.

-Discussion,

Not sure the need for 4th paragraph discussing the IL33/ST2 axis in renal cell carcinoma as the current review is about diabetic kidney disease. The entire study needs to be omitted unless the authors want to discuss in ‘IL33/ST2 and other kidney diseases’ section.

Same applies to 5th paragraph.

Thank You for both comments. We moved the 4th and 5th paragraphs into a new, short sub-section (4.2), named “IL-33/ST2 in other renal diseases”, as You proposed.

-Conclusions,

The authors wrote ‘Despite the relatively low number of studies published to date, the link between IL-33/ST2 axis and renal as well as diabetic kidney disease appears to be more and more clear’. Very confusing. What do they mean by renal as well as diabetic kidney disease?

Thank You. Our mistake. We rephrased the sentence into “Despite the relatively low number of studies published to date, the link between IL-33/ST2 axis and diabetic kidney disease appears to be more and more clear”.

I suggest the authors have the manuscript reviewed by a nephrologist for proper language and corrections.

Thank You. The manuscript was revised, as You correctly suggested.

B. Minor comments:

Introduction:

Discuss the distribution of IL33/ST2 in Kidney. And if the literature is available to discuss the location of the IL33/T2 axis in the renal tissue and pathological involvement of the diabetic kidney disease. For example, Chen et al. found that expression of ST2 is upregulated in human diabetic kidney tissues, (5).

Thank You. We shortly expanded the Introduction section adding evidence towards the IL-33 expression in the kidney.

Page 5 of 8.

Line 98. Authors wrote ‘A larger cohort (238 patients with CKD) was enrolled in an observational study conducted in Turkey some years later [28]. Reference 28 is wrong. Should be 25.

Correct, our mistakes. We modified accordingly.

Page 5 of 8,

Line 131, Under Studies on other renal diseases:

Consider discussing, the study by Mok et al., (1) on the IL33/ST2 axis in SLE and Lupus nephritis. Authors found that elevated serum ST2 level did not discriminate between active lupus nephritis and non-renal lupus. And about the IL33/ST2 in Acute kidney injury, (2) and in renal transplantation, (3,4).

Thank You. In the Discussion section, we added the works suggested under the sub-paragraph “4.2 IL-33/ST2 in other renal diseases”. We omitted them in the Results section for coherence with the main focus of the paper.

References:

1. Mok, M.Y.; Huang, F.P.; Ip, W.K.; Lo, Y.; Wong, F.Y.; Chan, E.Y.; Lam, K.F.; Xu, D. Serum levels of IL-33 and soluble ST2 and their association with disease activity in systemic lupus erythematosus. Rheumatology 2010, 49, 520–527

2. Akcay, A.; Nguyen, Q.; He, Z.; Turkmen, K.; Won Lee, D.; Hernando, A.A.; Altmann, C.; Toker, A.; Pacic, A.; Ljubanovic, D.G.; et al. IL-33 exacerbates acute kidney injury. J. Am. Soc. Nephrol. 2011, 22, 2057–2067.

3. Zhang, J.; Wang, Z.; Xu, Z.; Han, Z.; Tao, J.; Lu, P.; Huang, Z.; Zhou, W.; Zhao, C.; Tan, R.; et al. The potential role of IL-33 in renal transplant recipients with chronic allograft dysfunction. Ann. Transpl. 2016, 21, 611–618.

4. Mansell, H.; Soliman, M.; Elmoselhi, H.; Shoker, A. Elevated circulating Interleukin 33 levels in stable renal transplant recipients at high risk for cardiovascular events. PLoS ONE 2015, 10, e0142141.

5. Chen, N.K.; Chong, T.W.; Loh, H.L.; Lim, K.H.; Gan, V.H.; Wang, M.; Kon, O.L. Negative regulatory responses to metabolically triggered inflammation impair renal epithelial immunity in diabetes mellitus. J. Mol. Med. 2013, 91, 587–598.

Reviewer 2 Report

Over all, I would suggest to improve the English editing throughout the manuscript.

I did not see any PRISMA (or any other similar) guidelines followed in this manuscript.

What is the rationale of this manuscript? Its not clear in the introduction. I am assuming it's a pathophysiological outcomes of IL33/ST2 axis.

Replace Materials and Methods with Methods.

Table-2: Give information about sample characteristics, such as age, ethnicity and sex.

Please re-write the discussion section. What is the public health / therapeutic intervention aspects of IL-33 / ST2 axis. Do we have any therapeutic target?

Author Response

Over all, I would suggest to improve the English editing throughout the manuscript.

Thank You. The manuscript was carefully revised as suggested.

I did not see any PRISMA (or any other similar) guidelines followed in this manuscript.

Thank You. In fact, this is not a systematic literature review, but it is rather a descriptive brief review. We decided to not follow the PRISMA guidelines as in this specific topic too little works have been published to date, making the use of PRISMA difficult and somewhat misleading.

What is the rationale of this manuscript? Its not clear in the introduction. I am assuming it's a pathophysiological outcomes of IL33/ST2 axis.

Thank You. Effectively, this is the rationale of the present work and was now specified at the end of the Introduction section.

Replace Materials and Methods with Methods.

Thank You. We corrected accordingly.

Table-2: Give information about sample characteristics, such as age, ethnicity and sex.

Thank You. We added a dedicated column for describing such information.

Please re-write the discussion section. What is the public health / therapeutic intervention aspects of IL-33 / ST2 axis. Do we have any therapeutic target?

Thank You for Your question. At first, the Discussion section was now modified by inserting two sub-paragraph, the first of which dealing with “IL-33/ST2 in diabetic kidney disease” (4.1), and the second about “IL-33/ST2 in other renal diseases” (4.2). Since additional references have been added, we also had the opportunity to include some interesting cues arising from such studies, including the possible therapeutic target, represented by IL-33 in the AKI, as observed by Akcay and colleagues in one of the newly cited works. In addition, possible use of IL-33/ST2 axis as a target for a tailored treatment aimed at decreasing the burden of this specific condition was hypothesized, as already happens with other conditions, including inflammatory disorders.

Reviewer 3 Report

The review topic is interesting. The literature search is complete and thorough. The review manuscript is well written and clearly explain complex content. The figures are nicely illustrated. 

Author Response

The review topic is interesting. The literature search is complete and thorough. The review manuscript is well written and clearly explain complex content. The figures are nicely illustrated.

Thank You for Your kind words.

Round 2

Reviewer 1 Report

I have reviewed the original manuscript and made comments and suggestions.

The authors have answered to my comments and significantly modified the article.

Author Response

Thank You for Your kind comments.